# Features of the Oxidation of Multilayer (TiAlCrSiY)N/(TiAlCr)N Nanolaminated PVD Coating during Temperature Annealing

**Anatoly Ivanovich Kovalev [1,*]**, **Vladimir Olegovich Vakhrushev [1,2]**, **Egor Pavlovich Konovalov [1]**,
**German Simonovich Fox-Rabinovich [3]**, **Dmitry Lvovich Wainstein [1]**, **Stanislav Alekseevich Dmitrievskii [1]**
**and Alise Denisovna Mukhsinova [4]**

[1]  State Scientific Centre, I.P. Bardin Central Research Institute for Ferrous Metallurgy, 23/9 Bdg. 2, Radio Str., 105005 Moscow, Russia
[2]  MIREA-Rusian Technological University, 78 Vernadsky Avenue, 113454 Moscow, Russia
[3]  Department of Mechanical Engineering (JHE-316), McMaster University, 1280 Main Street West, Hamilton, ON L8S 4L7, Canada
[4]  Department of Chemistry and Technology of Crystals, Mendeleev University of Chemical Technology, 9 Miusskaya Square, 125047 Moscow, Russia
*  Correspondence: a_kovalev@sprg.ru

**Abstract:** A nano-multilayer $Ti_{0.2}Al_{0.55}Cr_{0.2}Si_{0.03}Y_{0.02}N/Ti_{0.25}Al_{0.65}Cr_{0.1}N$ PVD coating was deposited on Kennametal carbide K 313 inserts. These coatings are widely used to protect cutting tools under severe exploitation conditions. Under equilibrium conditions, it was found that the $Al_2O_3$ oxide possessed better adhesive properties than the $TiO_2$. The addition of chromium further enhanced the oxidation resistance of the coatings. Silicon significantly increased the oxidation resistance of this type of coating. The properties of the diffusion process in this coating have not been sufficiently investigated, despite the considerable number of articles published on this topic. For the purpose of this study, a multilayer ion-plasma (TiAlCrSiY)N/(TiAlCr)N coating was oxidized under equilibrium conditions; its chemical inhomogeneity was studied by time-of-flight mass spectroscopy using a TOF SIMS5-100 instrument. The data was collected from an area of $100 \times 100$ μ. A D-300 profilometer (KLA-Tencor Corp., Milpitas, California 95035, USA) was used to determine the rate of ion etching. It was found that oxidation commenced at the surface nanolayer of a TiAlCrN nitride, forming loose films of $Cr_2O_3$, $TiO_2$, and $Al_2O_3$ oxides. This passivating film had a thickness of around 140 nm. For the first time, the interlayer diffusion coefficients of Si and Y were determined in multilayer coatings based on $Ti_{0.2}Al_{0.55}Cr_{0.2}Si_{0.03}Y_{0.02}N/Ti_{0.25}Al_{0.65}Cr_{0.1}N$, under open air annealing at 700 °C. The physical nature of the differences in the diffusion of these elements is discussed. The diffusion rate in the near-surface volumes was lower than in the deep layers of the multilayer coating, most likely due to the formation of passivating oxide films on the surface.

**Keywords:** multilayer wear-resistant TiAlCrSiYN/TiAlCrN coatings; planar plasmon coatings; diffusion of thin layers; SIMS; oxide films

## 1. Introduction

Mono- and multilayer TiAlCrN-based coatings with inclusions of Si and Y have been intensively studied for machining applications. These wear-resistant coatings have high hardness, wear resistance, and thermal barrier properties [1].

The oxidation of such coatings deserves attention for several reasons: The first is due to the need for protective, heat-resistant materials—including cutting tools that operate under aggressive external conditions. In this case, the coatings form oxide films, which serve as diffusion barriers against aggressive chemical elements that cause metal degradation [2].

Such coatings extend the cutting life of a tool due to the dynamic formation of tribo-ceramics within the wear area at high temperatures of 700–1000 °C under conditions of dry, high-speed cutting [3].

The inclusion of several elements in the chemical composition of basic nitrides (TiN, TiAlN, TiCrAlN) in multilayer coatings is designed to significantly increase the service life of cutting tools. High hardness and oxidation resistance, combined with low thermal conductivity and coefficients of friction, are the main requirements for effective coatings applied on cutting tools that operate under extreme cutting conditions; this is why there exist a high variety of state-of-the-art, complex, alloyed, wear-resistant coatings based on the traditional TiCrAlN nitride. In each of these nitride coatings, oxidation plays a useful role in the protection of cutting tools under exploitation. It has been found that a coating with an inclusion of aluminum of up to 0.6 atomic% increases the temperature of oxide formation by 400 °C [4], compared to a TiN coating. Under equilibrium conditions, the $Al_2O_3$ oxide has better adhesive properties than $TiO_2$. The addition of chromium further enhances the oxidation resistance of coatings [5].

Cr-rich and Cr/Al oxides appear uniformly across coating surfaces during the oxidation of complex, nitride-based coatings with chromium additives. They provide an effective barrier to the inward diffusion of oxygen [1,5,6]. The oxidation of coatings containing Al and Ta develops in a competitive way. At first, the formation of amorphous $Al_2O_3$ oxides is predominant, but once aluminum is depleted, tantalum starts to oxidize into crystalline $Ta_2O_5$. Both oxides have high hardness and low thermal conductivity [7]. There have also been positive experiences using nitride-based coatings with ytterbium or yttrium additives [8,9].

Silicon is the most effective element for increasing the oxidation resistance of such coatings [10,11]. The formation of refractory and hard oxides that occur under severe conditions of high-temperature friction and wear represent an area of interest for the development of complex coatings. The depth distribution of oxides is the main focus of studies that are concerned with the oxidation of multilayer, complex nitride coatings [12].

Despite the enormous number of research works on this issue, the physical properties of diffusion in such coatings have not received sufficient attention. In this study, the multilayer PVD (TiAlCrSiY)N/(TiAlCr)N coating underwent oxidation under equilibrium conditions. The oxidation of a multicomponent multilayer system with a large number of interfaces is a very complex physical process. In a coating based on nonequilibrium multicomponent nitrides, diffusion occurs through the exchange mechanism and the movement of atoms through vacancies.

During oxidation, the depth-ward drift of oxygen atoms is accompanied by interactions with nitride components, as well as the formation of two-dimensional oxides and their subsequent growth. These phase transformations are determined by the thermodynamics of oxidation reactions [13] and depend on the diffusion permeability of the newly formed oxides. The activation energy of boundary diffusion is usually much less than that of bulk diffusion. Diffusing atoms are reflected from the boundaries when the interfaces are perpendicular to the direction of diffusion; the rate of diffusion is thereby lessened.

Apart from oxygen, Si and Y are also diffused from the (TiAlCrSiY)N layer to the (TiAlCr)N layer due to the concentration gradient as well as heating during oxidation. A similar migration of W, Nb has been observed previously during the tribo-oxidation of multilayer coatings [14]. Any predictions in such an oxidative process are extremely difficult to make. An analysis of the experimental results of the interlayer distribution of Si and Y will reveal the diffusion pattern of oxidation in this type of complex coating. It is presumed that Si reduces the grain size, whereas yttrium inhibits grain growth at elevated temperatures, sets apart grain boundaries, blocks diffusion paths, and increases the stability of the fine-grained structure [14]. These elements are also believed to act in a synergistic way. The chief aim of this study is to investigate the features of Si and Y diffusion within a multilayer $Ti_{0.2}Al_{0.55}Cr_{0.2}Si_{0.03}Y_{0.02}N/Ti_{0.25}Al_{0.65}Cr_{0.1}N$ PVD coating.

## 2. Materials and Methods

A multilayer $Ti_{0.2}Al_{0.55}Cr_{0.2}Si_{0.03}Y_{0.02}N/Ti_{0.25}Al_{0.65}Cr_{0.1}N$ coating was deposited using $Ti_{0.2}Al_{0.55}Cr_{0.2}Si_{0.03}Y_{0.02}$ and $Ti_{0.25}Al_{0.65}Cr_{0.1}$ 1 targets, correspondingly fabricated by a powder metallurgical process on a cemented carbide cutting insert WC-Co substrate in a R&D-type hybrid PVD coater (Kobe Steel Ltd., Onoecho Ikeda, Kakogawa, Hyogo, Japan) using a plasma-enhanced arc source. The WC-Co samples were heated to 500 °C and cleaned by Ar-ion etching. During the PVD process, an Ar–$N_2$ mixture gas was fed into the chamber at a pressure of 2.7 Pa with a nitrogen partial pressure of 1.3 Pa. The arc source was operated at 100 A on a 100 mm diameter × 16 mm-thick target. The other deposition parameters were a bias voltage of 100 V and a substrate rotation of 5 rpm. The thickness of the studied coating was around 3.5 μm for the film characterization. The coating had a columnar nanocrystalline multilayered microstructure with alternating nanolayers at every 50 nm [15,16], a hardness of 30 GPa—according to the nanoindentation at room temperature—and 28 GPa at 500 °C [17].

The multilayer ion-plasma (TiAlCrSiY)N/(TiAlCr)N coating was oxidized under equilibrium conditions under open air annealing at 700 °C for 15, 30, and 45 min.

Chemical inhomogeneity was studied by time-of-flight mass spectroscopy on a TOF SIMS5-100 instrument with high sensitivity in the ppb range using the time-of-flight mass spectroscopy method. This device was equipped with a time-of-flight analyzer for positive and negative secondary ions with an EDR system for assessing low-intensity signals emitted during intense cascade flows of secondary ions. Two ion sources, coordinated in time, were used to study the depth distribution of the chemical elements. Ion etching was carried out with sources of $Cs^+$ or $O^+$ ions (0.5 keV), and a source of $Bi^+$ ions (25 keV) was used to excite secondary ions from the sample. Ion etching was carried out over an area of 300 × 300 μ at different rates, depending on the material studied. The analysis was carried out on an area of 100 × 100 μ. The profilometer D-300 device (KLA-Tencor Corp., USA) was used to determine the rate of the ion etching profilometer calibration on standard silicon samples, with an accuracy of up to 0.5 nm. The processing of the mass spectrometry data for determining the atomic and mass concentrations of elements was carried out using the SurfaceLab 7 program (IONTOF, Muenster, Germany).

## 3. Results

The coating had a complex structure that combined a nano-multilayered structure with a modulating composition with a columnar structure (with a grain size of around 5–20 nm).

Figure 1a presents a HR TEM image of the TiAlCrSiYN layer. The coating had an ultra-fine nano-grain size in as-deposited conditions. The grain size of the coating was within 5–10 nm.

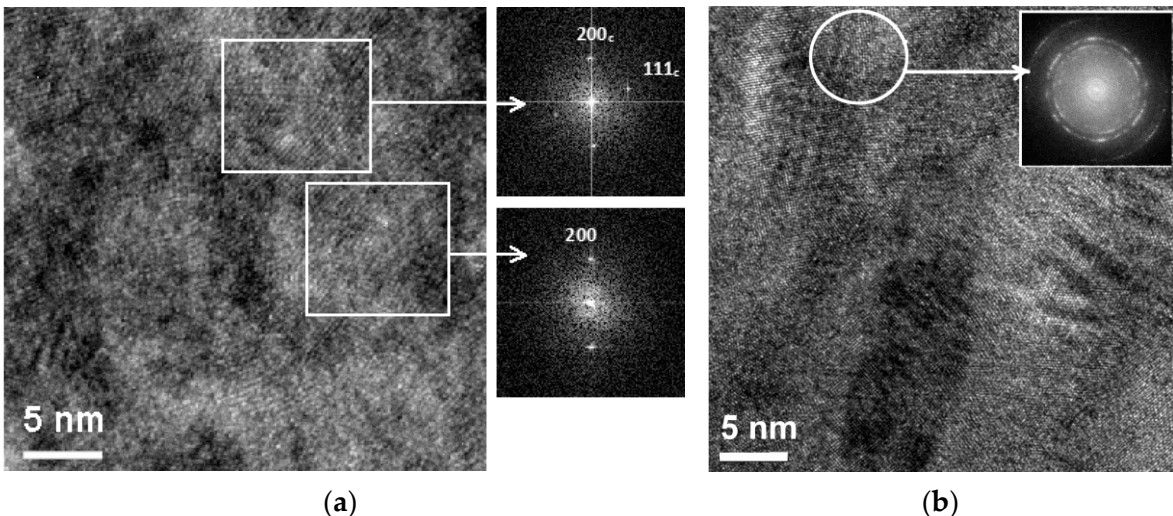

**(a)**                                                                                           **(b)**

**Figure 1.** TEM image of the (TiAlCrSiY)N layer (**a**) and the (TiAlCr) layer (**b**) of the nanolaminate coating.

Figure 1b presents a zero-loss HRTEM image of the coating. The image does not show epitaxial growth between the layers. The layers were polycrystalline and did not have a preferred orientation, as can be seen in the ring diffraction pattern.

After annealing the samples in an open furnace, a layer of oxides appeared on the surface of the coating, which had a porous, layered microstructure. Mass spectroscopy made it possible to establish the phase composition of the oxidized surface. Figure 2 shows that complex oxide films were formed on the sample surface during heat treatment; these were oxides of chromium and titanium. Similar phases appear under conditions of high-speed dry cutting [7]. Mullite-like (Al-Si-O) tribo-films with a high protective ability are also formed [8]. The depth distribution of chemical elements and phases was evaluated through cyclic etching of the surface with $Cs^+$ and $Bi^+$ ions. During active surface etching with cesium ions, layers of atomic thickness were sequentially removed. Etching with bismuth ions in the static mode during pauses excited a flow of scattered secondary ions, which were analyzed in a mass spectrometer. Such complex surface etching made it possible to represent the deep distribution of the chemical elements.

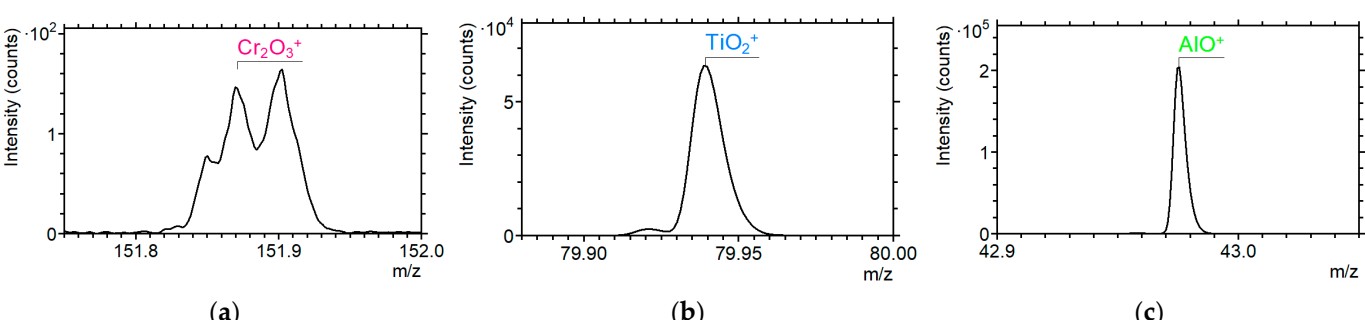

**(a)**                                                       **(b)**                                                       **(c)**

**Figure 2.** Mass spectra of secondary $Cr_2O_3^+$ (**a**), $TiO_2^+$ (**b**), and $AlO^+$ (**c**) ions from the TiAlCr-SiYN/TiAlCrN coating after annealing in an open furnace for 45 min at 700 °C, obtained at different depths from the surface during ion etching.

These data provide a basis for a model of the distribution of phases across the depth of the oxide film, presented in Figure 3. Si and Y were not found within the equilibrium oxide films. After a 45 min annealing process, the oxide film had a thickness of about 140 nm. The 1.5 of the first bilayer was oxidized. Computer processing of the data obtained in each layer during surface ion etching made it possible to reconstruct a 3D picture of the distribution of the volume phases.

The 3D distribution of the titanium, chromium, and aluminum oxides in Figure 4 shows that during annealing, multilayer protective films were formed on the surface of the material. $Cr_2O_3$ was formed first, followed by $TiO_2$, and, finally, $Al_2O_3$. Further studies on the redistribution of Si and Y were carried out on an equilibrium oxide film. The surface oxide was completely removed from the sample by ion etching to avoid the implantation of oxygen into the sample. Figure 5 shows the depth distributions of Si and Y in the initial state and after annealing for 45 min. The zero point on the depth axis in Figure 5 corresponds to a true distance of 140 nm from the sample surface.

Figure 5 shows the change in the depth distribution of silicon and yttrium before and after annealing in the atmosphere at a temperature of 700 °C. It can be seen that after prolonged annealing, the thickness of the layers enriched with silicon and yttrium increased. In addition, silicon—being more mobile and having a smaller atomic radius than yttrium—had a longer diffusion path. The difference in the half-width of the depth distribution peaks (FWHM) is presented in Figure 5. The features of the elemental depth distribution were directly related to interlayer diffusion under surface oxidation. The change in this characteristic was evaluated in all depth profiles after heating to 700 °C for 15, 30, and 45 min in air; this made it possible to calculate the diffusion rate in a multilayer coating. Table 1 presents the results, which enabled the diffusion characterization of Si and Y in a multilayer coating.

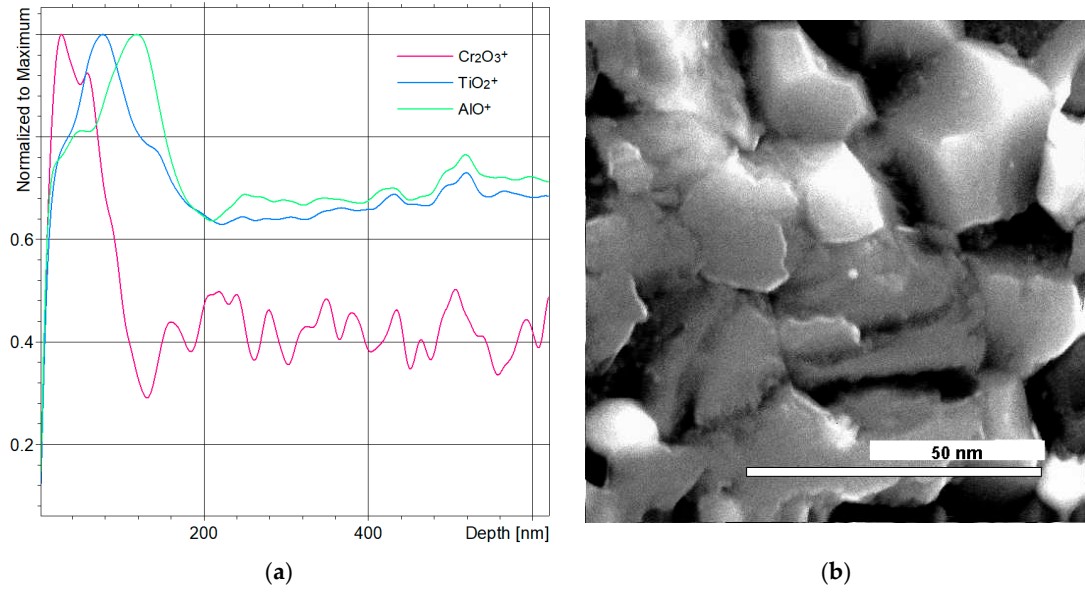

(**a**)    (**b**)

**Figure 3.** Depth profile of phase distribution in the oxide film on the TiAlCrSiYN/TiAlCrN coating after 45 min of annealing at a temperature of 700 °C (**a**). SEM image of the oxide surface (**b**).

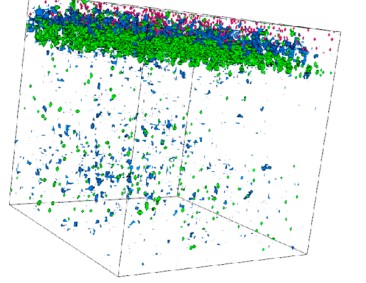

3D Render Overlay of: $Cr_2O_3^+$, $TiO_2^+$, $AlO^+$

**Figure 4.** A reconstructed 3D model of the TiAlCrSiYN/TiAlCrN coating for surface oxides after 45 min of annealing at 700 °C. (length of the X and Y axes is 50 μm, length of the Z axis is 670 nm).

**Table 1.** The change in the depth penetration, ΔX, of silicon and yttrium in the layers at the initial stage and after annealing at a temperature of 700 °C for $Ti_{0.2}Al_{0.55}Cr_{0.2}Si_{0.03}Y_{0.02}N/Ti_{0.25}Al_{0.65}Cr_{0.1}$.

| Time | N Peak | Y | | Si | |
| | | FWHM | | FWHM | |
| | | X, nm | ΔX, nm | X, nm | ΔX, nm |
|---|---|---|---|---|---|
| 0 min, RT | 1 | 35 | 0 | 35 | 0 |
| | 2 | 29 | 0 | 28 | 0 |
| | 3 | 35 | 0 | 36 | 0 |
| 15 min | 1 | 43 | 8 | 44 | 9 |
| | 2 | 37 | 8 | 36 | 8 |
| | 3 | 42 | 7 | 44 | 8 |
| 30 min | 1 | 48 | 13 | 51 | 16 |
| | 2 | 40 | 11 | 42 | 14 |
| | 3 | 44 | 9 | 49 | 13 |
| 45 min | 1 | 51 | 16 | 60 | 25 |
| | 2 | 44 | 15 | 57 | 29 |

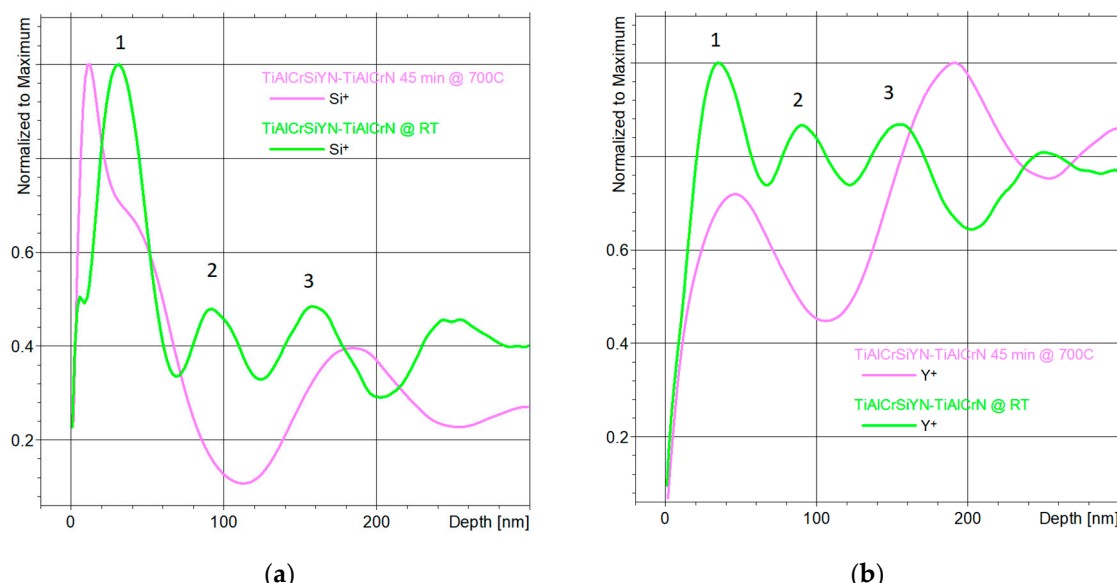

(**a**)                                    (**b**)

**Figure 5.** Depth distribution of silicon (**a**) and (**b**) yttrium on the surface of a $Ti_{0.2}Al_{0.55}Cr_{0.2}Si_{0.03}Y_{0.02}N/Ti_{0.25}Al_{0.65}Cr_{0.1}N$ coating before and after annealing at 700 °C for 45 min.

Formula (1) was used to calculate the diffusion rate and path for the studied multi-layer samples [18]:

$$C(x,t) = \frac{C_0}{2}\left(erf\frac{h-x}{2\sqrt{Dt}} + erf\frac{h+x}{2\sqrt{Dt}}\right) \qquad (1)$$

where $C_0$ is the initial concentration of elements in the layer, $t$ is the diffusion annealing time, $h$ is the thickness of the initial layer, and $\Delta x$ is the diffusion path during annealing.

This expression was used to determine the main characteristics of the interlayer diffusion of silicon and yttrium.

Table 2 shows that the diffusion rate in a multilayer coating depends on the number of bilayers. The fewer there are, the higher is the diffusion rate—which may be related to the mechanism of the process itself. The coatings have a nanometer thickness, which predetermines the excess surface energy of the nanomaterial. Therefore, according to thermodynamic principles, it is difficult to contain a large number of vacancies and other

defects within the volume of the nanomaterial. Such nanolaminate coatings differ from other, thicker ones obtained by magnetron sputtering on a micrometer scale [19].

**Table 2.** The relationship between silicon and yttrium diffusion rates in the layers and its effect on annealing time at a temperature of 700 °C in a multilayer [20]: coating.

| Time | N Peak | Y | | Si | |
|---|---|---|---|---|---|
| | | D, m$^2$/s | D Average, m$^2$/s | D, m$^2$/s | D Average, m$^2$/s |
| 15 | 1 | $8.4 \times 10^{-20}$ | | $1.3 \times 10^{-19}$ | |
| | 2 | $1.0 \times 10^{-19}$ | $8.9 \times 10^{-20}$ | $1.0 \times 10^{-19}$ | $1.1 \times 10^{-19}$ |
| | 3 | $7.9 \times 10^{-20}$ | | $1.0 \times 10^{-19}$ | |
| 30 | 1 | $1.4 \times 10^{-19}$ | | $2.1 \times 10^{-19}$ | |
| | 2 | $9.8 \times 10^{-20}$ | $1.0 \times 10^{-19}$ | $1.6 \times 10^{-19}$ | $1.7 \times 10^{-19}$ |
| | 3 | $6.5 \times 10^{-20}$ | | $1.4 \times 10^{-19}$ | |
| 45 | 1 | $1.4 \times 10^{-19}$ | | $3.4 \times 10^{-19}$ | |
| | 2 | $1.2 \times 10^{-19}$ | $1.3 \times 10^{-19}$ | $4.5 \times 10^{-19}$ | $3.9 \times 10^{-19}$ |

The scheme of Figure 6 helps us to understand in which directions silicon and yttrium are diffused. Oxidation begins from the surface nanolayer of a TiAlCrN nitride and forms a loose film of $Cr_2O_3$, $TiO_2$, and $Al_2O_3$ oxides. This passivating film has a thickness of around 140 nm; its role is to inhibit the further-depth penetration of oxygen into the multilayer nitride coating. Oxygen blocks the boundaries of columnar crystals in the lower bilayer and affects diffusion. Following deposition, the coating structure is columnar. Due to this, silicon and yttrium diffusion can proceed both over the grain volume and along the grain boundaries [14]. Grain boundary diffusion should accelerate mass transfer; however, the diffusion rate is lower in the bilayer underneath the oxide, increasing at greater depths. From this, it can be concluded that the boundaries of columnar crystals in the first bilayers are deactivated by oxygen penetrating into the depths of the coating layer. In the lower layers of the coating, the diffusion of Si and Y depends on the crystal structure of both nitrides. The difference between the diffusion rates of Si and Y in these volumes indicates that diffusion does not depend on the improper orientation of the grains [19], but is determined only by the size of the atoms of these elements.

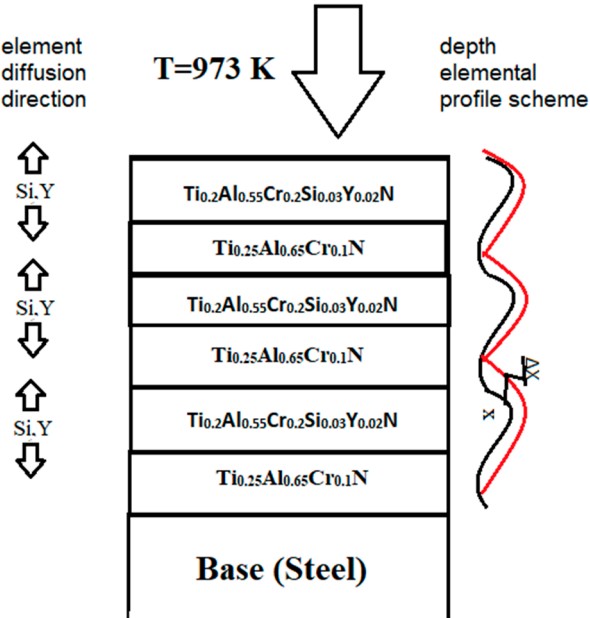

**Figure 6.** Scheme of diffusion occurring during atmospheric annealing at a temperature of 700 °C in the TiAlCrSiYN/TiAlCrN multilayer coating. On the right is a diagram of the depth distribution of the diffusing elements (Si, Y). The black line indicates the depth profile before heating; the red line represents the depth profile (Si, Y) after annealing; ΔX is the diffusion path of atoms (Si, Y).

## 4. Conclusions

1.  The coefficients and rates of diffusion of the chemical elements of multilayer thin films were calculated on the basis of their depth distribution profiles. Secondary ion mass spectroscopy was used to obtain depth profiles with a high sensitivity and depth resolution.
2.  The effective diffusion coefficients of silicon and yttrium were determined in a multilayer nanolaminate $Ti_{0.2}Al_{0.55}Cr_{0.2}Si_{0.03}Y_{0.02}N/Ti_{0.25}Al_{0.65}Cr_{0.1}N$ coating after annealing at a temperature of 700 °C in air.
3.  It had been established that the diffusion rate of Si was several times-higher than that of Y, which can be associated with a significant (about two times) difference in the sizes of the atoms between these elements.
4.  It had been found that the diffusion rate in near-surface volumes was lower than in the deep layers of a multilayer coating—most likely due to the formation of passivating oxide films on the surface.
5.  An anomaly of the preferential diffusion of Si in at-surface volumes has been established; this may be due to the lower stability of metal-nitride atomic bonds and the greater affinity of Si for oxygen compared to Y. These assumptions require their experimental or theoretical verification in the future.

**Author Contributions:** A.I.K.—conceptualization, validation and methodology, V.O.V.—formal analysis, investigation, E.P.K.—investigation, writing—original draft preparation, G.S.F.-R.—supervision, writing—review and editing, D.L.W.—data curation, writing—review and editing, S.A.D.—data curation, writing—original draft preparation, A.D.M.—visualization, investigation. All authors have read and agreed to the published version of the manuscript.

**Funding:** The investigation was supported by the Russian Science Foundation, Grant No. 21-79-10044.

**Institutional Review Board Statement:** Not applicable.

**Informed Consent Statement:** Not applicable.

**Data Availability Statement:** Raw experimental data will be available on reasonable request.

**Acknowledgments:** A critical proofreading of the manuscript by Michael Dosbaev is acknowledged.

**Conflicts of Interest:** The funders had no role in the design of the study; in the collection, analyses, or interpretation of data; in the writing of the manuscript; or in the decision to publish the results.

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
