# Peer review of "Features of the Oxidation of Multilayer (TiAlCrSiY)N/(TiAlCr)N Nanolaminated PVD Coating during Temperature Annealing"

_coatings, doi:10.3390/coatings13020287_

Round 1

Reviewer 1 Report

1.      Abstract should be given as more interesting. Express at least one of the main aspects and features of the paper.

2.      Improve the conclusion part of the Abstract.

3.      Wherever applicable, the scientific explanation needs to be added and the research novelties need to be clearly emphasized.

4.      At the end of Introduction section, it would be better to add the paper's organization in different sections.

5.      Given that the manuscript is based on experimental work and measurements, it is vital for the authors to report on the method(s) to improve measurement reliability. The methods/measures that have been taken to minimize experimental errors and improve reliability should be included in the experimental setup.

6.      Further, results and analysis of experiments should be compared with previous researchers by citing references.

7.      Improve the quality of image in Figure 4.

8.      Provide citation of literature for Eq. (1).

9.      Page 5, Line 162 provide the correct information of chemical formula.

10.  Improve the conclusion with scope for future work.

11.  Conclusion must be presented in highlight the contribution, and applicability of the work.

12.  Please check the manuscript for wrong choice of words, grammatical errors and incoherent sentence structure.

Author Response

Author's Notes to Reviewer 1

  1. Abstract should be given as more interesting. Express at least one of the main aspects and features of the paper.

Annotation expanded:

These coatings are widely used to protect cutting  tools under severe exploitation conditions. For the first time, the interlayer diffusion coefficients of Si and Y were determined in  multilayer coatings based on Ti0.2Al0.55Cr0.2Si0.03Y0.02N / Ti0.25Al0.65Cr0.1N, under open air annealing at 700°C. The physical nature of the differences in the diffusion of these elements is discussed.

  1. Improve the conclusion part of the Abstract.

Annotation expanded:

hese coatings are widely used to protect cutting  tools under severe exploitation conditions. For the first time, the interlayer diffusion coefficients of Si and Y were determined in  multilayer coatings based on Ti0.2Al0.55Cr0.2Si0.03Y0.02N / Ti0.25Al0.65Cr0.1N, under open air annealing at 700°C. The physical nature of the differences in the diffusion of these elements is discussed.

And the conclusion #5 was added

  1. An anomaly of preferential diffusion of Si in at surface volumes has been established. This may be due to the lower stability of metal-nitride atomic bonds and the greater affinity of Si for oxygen compared to Y. These assumptions require their experimental or theoretical verification in the future.
  2. Wherever applicable, the scientific explanation needs to be added and the research novelties need to be clearly emphasized.

Additions have been made to the text of the annotation and Results.

  1. At the end of Introduction section, it would be better to add the paper's organization in different sections.

The plan is usually presented in the introduction to the report or theses. In this case, the authors use the generally accepted standard structure of the article.

  1. Given that the manuscript is based on experimental work and measurements, it is vital for the authors to report on the method(s) to improve measurement reliability. The methods/measures that have been taken to minimize experimental errors and improve reliability should be included in the experimental setup.

Phrases about the sensitivity of the time-of-flight mass sectroscopy method and the accuracy of the profilometry method using certified standards have been added to the text of the article.

Chemical inhomogeneity was studied by time-of-flight mass spectroscopy on a TOF SIMS5-100 instrument high sensitivity in the ppb range of the time-of-flight mass spectroscopy method

The profilometer D-300 device (KLA-Tencor Corp., USA) was used to determine the rate of ion etching profilometer calibration on standard silicon samples with an accuracy of up to 0.5 nm

  1. Further, results and analysis of experiments should be compared with previous researchers by citing references.

Unfortunately, data on Si and Y interlayer diffusion in such complexly organized multilayer nanolaminate coatings have been obtained for the first time and they are not available in any publications. This is the novelty of this work.

  1. Improve the quality of image in Figure 4.

Figure 4 is an original computer 3D reconstruction of elemental distribution maps during layer-by-layer ion etching. These are experimental data and its cannot be artificially improved. The data in Figure 4 very well correlates with the graphs in Figure 3.

  1. Provide citation of literature for Eq. (1).

The literary reference to equation (1) has been added to the list of references.

  1. Page 5, Line 162 provide the correct information of chemical formula.

The information has been checked and is correct.

  1. Improve the conclusion with scope for future work.

The conclusion #5 was added

  1. An anomaly of preferential diffusion of Si in at surface volumes has been established. This may be due to the lower stability of metal-nitride atomic bonds and the greater affinity of Si for oxygen compared to Y. These assumptions require their experimental or theoretical verification in the future.

  1. Conclusion must be presented in highlight the contribution, and applicability of the work.

The conclusion #6 was added

The obtained results are of great importance for understanding the stability of multicomponent nonequilibrium nitrides during high-temperature oxidation. The studied multilayer nitride coatings on cutting tools are very promising for the machining of steels and heat-resistant alloys. In the cutting zone, the phenomenon of tribo-oxidation is always observed. And these obtained results will help to interpret the physical mechanism of tribooxidation.

  1. Please check the manuscript for wrong choice of words, grammatical errors and incoherent sentence structure.

The text has been edited

Reviewer 2 Report

In this paper, the authors have studied the oxidation of (TiAlCrSiY)N/(TiAlCr)N coatings at 700 °C in air. The duration of the experiment was 45 min. The work is interesting and novel. However, the discussion of the results is confusing. Essential results - cross-section images before and after oxidation – are missing in the manuscript. The following comments should be addressed:

1.You need to use subscripts for all oxides in the paper. Write Al2O3 instead of Al2O3, TiO2 instead of TiO2, etc. The same applies to nitrides.

2.Oxidation conditions (air, 700 °C, 45 min) should be clearly specified in the methods section. It is not enough to mention them in the abstract only.

3.The authors stated that the coating had 3 bilayers (line 104). Furthermore, they said that the thickness of each layer was 50 nm (line 111). However, there is no TEM/SEM image of the cross-section in the paper to confirm it. It is necessary to show the cross-section of the multilayer film in the article.

4.Fig.3 (SIMS depth profile) shows that the thickness of the oxide scale was ~200 nm. Does it mean that the first two bilayers have been fully oxidized? Please, explain.

5.The discussion of the results is confusing. Fig. 6 (a schematic) suggests that only a top layer has been oxidized, which means that it had more than 150 – 200 nm. If the top layer was thicker, you need to explain it. Furthermore, it is necessary to show the cross section of the oxidized material (SEM/TEM) in addition to the schematic.

6.Fig. 5 shows the distribution of Si and Y before and after oxidation. The concentration of Si in the oxidized material was higher at the surface. The opposite was found for Y. It means that Si was preferentially diffusing towards the surface of the film and Y towards the bulk. Do you have any explanation for this behavior?

7.The diffusion fluxes of Si and Y were apparently in opposite directions as comes from results in Fig. 5. You need to indicate it in the schematic (Fig. 6) as well.

8.It would also help to include the curves after 15 and 30 min in Fig. 5 to show the dynamics of diffusion. The authors apparently measured them (see the data in Table 1).

9.Decimal points and superscripts should be used in Table 2. For example, 8,4E-20 should be rewritten as 8.4x10-20, etc.

10.The discussion of the grain structure of the films is also confusing. At the beginning, the authors said that the layers were polycrystalline and without any preferential orientation (line 134). In the end, however, they say that after magnetron sputtering the coating structure became columnar (line 218). You need to show the grain structure. Was it columnar or polycrystalline? Furthermore, you need to clearly explain what happened with the materials and why the grain structure has changed.

Author Response

Author's Notes to Reviewer 2

Many thanks to the reviewers for a careful reading of the article and very useful comments.

1.You need to use subscripts for all oxides in the paper. Write Al2O3 instead of Al2O3, TiO2instead of TiO2, etc. The same applies to nitrides.

All phases are indexed.

2.Oxidation conditions (air, 700 °C, 45 min) should be clearly specified in the methods section. It is not enough to mention them in the abstract only.

The “The multilayer ion-plasma (TiAlCrSiY)N / (TiAlCr)N coating was oxidized under equilibrium conditions under open air annealing at 700°C for 15, 30 and 45 minutes.” Was added in

  1. Materials and Methods

3.The authors stated that the coating had 3 bilayers (line 104). Furthermore, they said that the thickness of each layer was 50 nm (line 111). However, there is no TEM/SEM image of the cross-section in the paper to confirm it. It is necessary to show the cross-section of the multilayer film in the article.

TEM of multilayer (TiAlCrSiY)N/(TiAlCr)N have been repeatedly cited in our previously published articles [G. Fox-Rabinovich, A. Kovalev, S. Veldhuis, K. Yamamoto, J. L. Endrino, I. S. Gershman, A. Rashkovskiy, M. H. Aguirre & D. L. Wainstein. Spatio-temporal behaviour of atomic-scale tribo-ceramic films in adaptive surface engineered nano-materials. Sci. Rep. 5 (2015), 8780; DOI:10.1038/srep08780; A. Kovalev, D. Wainstein, G. Fox-Rabinovich, S. Veldhuis, K. Yamamoto. Features of self-organization in nanostructuring PVD coatings on base of polyvalent metal nitrides under severe tribological conditions. Surface and Interface Analysis, 2008, vol. 40, issue 3-4, 881-884]

The architecture of multilayer coating is informatively presented in Figure 5-a.

The initial multilayer structure of the coating is presented in a different way. Figure 5 shows the distribution of silicon and yttrium in the initial structure. This information is unique and demonstrates the precision and high elemental sensitivity of time-of-flight mass spectroscopy for the study of multilayer coatings. In this case, the depth distribution of impurity elements with a concentration of 0.02-0.03 at% in complex nitrides is presented..

4.Fig.3 (SIMS depth profile) shows that the thickness of the oxide scale was ~200 nm. Does it mean that the first two bilayers have been fully oxidized? Please, explain.

The 1.5 of the first bilayer are oxidized. The thickness of the oxide on the surface of the coating is about 140 nm .This phrase has been added to the text. The Fig3-b added to Figure 3 is a general optical image of the ion etch crater.

5.The discussion of the results is confusing. Fig. 6 (a schematic) suggests that only a top layer has been oxidized, which means that it had more than 150 – 200 nm. If the top layer was thicker, you need to explain it. Furthermore, it is necessary to show the cross section of the oxidized material (SEM/TEM) in addition to the schematic.

The oxidized layer had a thickness of 140 nm. This was established by time-of-flight mass spectroscopy and atomic force profilometry. Added to Figure 3 is an optical microscopy view of the etch crater. At the moment, we cannot show the cross section using the TEM method. Samples for this must be destroyed. And now the phase composition of complex oxides is studied by X-ray photoelectron spectroscopy.

6.Fig. 5 shows the distribution of Si and Y before and after oxidation. The concentration of Si in the oxidized material was higher at the surface. The opposite was found for Y. It means that Si was preferentially diffusing towards the surface of the film and Y towards the bulk. Do you have any explanation for this behavior?

Thank you very much for this great tip. With your permission, we have added the following to the text of the article: «Fig. 5 shows the distribution of Si and Y before and after oxidation. The concentration of Si in the oxidized material was higher at the surface. The opposite was found for Y. It means that Si was preferentially diffusing towards the surface of the film and Y towards the bulk. This may be due to the lower stability of the metal-nitride atomic bonds and the greater affinity of Si for oxygen compared to Y.”

However, these assumptions require their experimental or theoretical verification in the future. The strength of interatomic metal-nitride bonds can be characterized by photoelectron spectroscopy for coatings in the initial state. In this case, the chemical shifts of the Si-N and Y-N characteristic lines in the photoelectron spectra are investigated. The affinity for the oxidation of Si and Y in the composition of a complex nitride can be qualitatively established by theoretical quantum calculations. These studies are expected in the future.

7.The diffusion fluxes of Si and Y were apparently in opposite directions as comes from results in Fig. 5. You need to indicate it in the schematic (Fig. 6) as well.

In the scheme of Figure 6, an arrow is added for the diffusion of elements towards the surface of the coating.

8.It would also help to include the curves after 15 and 30 min in Fig. 5 to show the dynamics of diffusion. The authors apparently measured them (see the data in Table 1).

Naturally, all curves are available, but the addition of these graphs can significantly complicate the presentation of experimental results and the number of figures. All the necessary data are presented in the tables.

9.Decimal points and superscripts should be used in Table 2. For example, 8,4E-20 should be rewritten as 8.4x10-20, etc.

These corrections are included in the tables

10.The discussion of the grain structure of the films is also confusing. At the beginning, the authors said that the layers were polycrystalline and without any preferential orientation (line 134). In the end, however, they say that after magnetron sputtering the coating structure became columnar (line 218). You need to show the grain structure. Was it columnar or polycrystalline? Furthermore, you need to clearly explain what happened with the materials and why the grain structure has changed.

The coating has a columnar nanocrystalline structure. Such a structure is quite typical for magnetron ion-plasma sputtering. The presence of a columnar structure does not suggest single crystallinity. The columnar elements of the structure are separated by high-angle boundaries, and inside they have a nanocrystalline structure. We did not study changes in the grain structure of the coating during heat treatment. The structural changes of the coating in the cutting area, we have repeatedly investigated and described in previous publications. Amorphization and dynamic recrystallization take place there. But these questions lie outside the scope of the study of interlayer diffusion under equilibrium conditions, that is, this article.

Round 2

Reviewer 1 Report

Authors have made significant changes and corrections in the revised manuscript. Thus, consider the manuscript for publication in its present form. Accepted from myside. 

Author Response

Many thanks to the reviewer for their attentive attitude to our article.

Reviewer 2 Report

The authors partially answered my previous comments in the cover letter. The manuscript is acceptable for publication subject to minor revision.

1.You need to include a caption of Fig. 3b in the paper (the bottom of the crater).

2.Fig. 4 (a 3D model of the distribution of oxides in the coating) must include a scale bar.

Author Response

Many thanks to the reviewer for their attentive attitude to our article. We have made all the necessary additions to the captions for Figures 3 and 4.

Figure 3. Depth profile of phase distribution in the oxide film on the TiAlCrSiYN/TiAlCrN coating after 45 minutes of annealing at a temperature of 700 °C(a).SEM iamage of oxide surface (b)

Figure 4. A reconstructed 3D model of the TiAlCrSiYN/TiAlCrN coating for surface oxides after 45 minutes of annealing at 700°C. (length of X and Y axes is 50 mm, length of Z axes is 670 nm)